# Phenotypic but not genetically predicted heart rate variability associated with all-cause mortality

Balewgizie S. Tegegne[1,10], M. Abdullah Said [2,10], Alireza Ani [1,3], Arie M. van Roon[4], Sonia Shah [5,6], Eco J. C. de Geus [7], Pim van der Harst[8], Harriëtte Riese[9], Ilja M. Nolte [1] & Harold Snieder [1✉]

Low heart rate variability (HRV) has been widely reported as a predictor for increased mortality. However, the molecular mechanisms are poorly understood. Therefore, this study aimed to identify novel genetic loci associated with HRV and assess the association of phenotypic HRV and genetically predicted HRV with mortality. In a GWAS of 46,075 European ancestry individuals from UK biobank, we identified 17 independent genome-wide significant genetic variants in 16 loci associated with HRV traits. Notably, eight of these loci (*RNF220, GNB4, LINCR-002, KLHL3/HNRNPA0, CHRM2, KCNJ5, MED13L*, and *C16orf72*) have not been reported previously. In a prospective phenotypic relationship between HRV and mortality during a median follow-up of seven years, individuals with lower HRV had higher risk of dying from any cause. Genetically predicted HRV, as determined by the genetic risk scores, was not associated with mortality. To the best of our knowledge, the findings provide novel biological insights into the mechanisms underlying HRV. These results also underline the role of the cardiac autonomic nervous system, as indexed by HRV, in predicting mortality.

[1] Department of Epidemiology, University of Groningen, University Medical Center Groningen, Groningen, The Netherlands. [2] Department of Cardiology, University of Groningen, University Medical Center Groningen, Groningen, The Netherlands. [3] Department of Bioinformatics, Isfahan University of Medical Sciences, Isfahan, Iran. [4] Department of Vascular Medicine, University of Groningen, University Medical Center Groningen, Groningen, The Netherlands. [5] Institute for Molecular Bioscience, The University of Queensland, Brisbane, QLD, Australia. [6] Institute of Cardiovascular Science, University College London, London, UK. [7] Department of Biological Psychology, Vrije Universiteit Amsterdam, Amsterdam, The Netherlands. [8] Department of Cardiology, Division of Heart and Lungs, University Medical Center Utrecht, University of Utrecht, Utrecht, The Netherlands. [9] Department of Psychiatry, Interdisciplinary Center Psychopathology and Emotion Regulation, University of Groningen, University Medical Center Groningen, Groningen, The Netherlands. [10] These authors contributed equally: Balewgizie S. Tegegne, M. Abdullah Said. ✉email: h.snieder@umcg.nl

Heart rate variability (HRV), a result of complex interactions between intrinsic cardiovascular regulatory mechanisms and extrinsic environmental factors, is a specific marker of parasympathetic (or vagal nerve) control of the heart rhythm. This physiological phenomenon was widely reported back in the 1990s as a predictor for increased cardiovascular and all-cause mortality[1–3]. Later, Zulfiqar and colleagues[4] showed that a high HRV was associated with longevity, indicating that preservation of autonomic function might be important to healthy survival into old age. However, the molecular mechanisms underlying the association between low HRV, and increased mortality are unknown.

There is consistent evidence for the influence of genetic factors on HRV[5] with heritability estimates ranging from 14 to 71%[5,6]. However, very few studies have tried to identify the genetic variants responsible for this heritability. Previous studies have attempted to associate variants in candidate genes based on current knowledge of parasympathetic nervous system biology, though the results could not be replicated[7]. In addition, three genome-wide association studies (GWASs) were performed to identify genetic variants associated with HRV. The first was conducted in the Framingham Heart Study that found no genome-wide significant results[8]. Nolte et al. conducted the second one, which detected 17 genome-wide significant genetic variants in eight loci[9]. While the previous two studies were conducted among European ancestries, a third GWAS of HRV in Hispanic/Latino individuals reported two genome-wide significant single-nucleotide polymorphisms (SNPs)[10] It was estimated that the identified common variants explained only up to 2.6% of the phenotypic variability of HRV, leaving a large proportion of heritability unexplained.

However, large cohorts such as the UK Biobank were not included in the reported GWASs, and only a limited number of SNPs imputed against the HapMap2 reference panel were analyzed. GWASs using larger cohorts combined with higher resolution SNP arrays and more accurate imputation reference datasets are expected to explain a larger part of the missing heritability of HRV. Identification of more genetic loci and underlying functional variants could help to not only elucidate the biological pathways through which reduced HRV contributes to cardiovascular disease risk but also facilitate investigation of its potential causal role in health outcomes. Therefore, this study aimed to: (1) identify novel genetic loci associated with HRV, and (2) assess the association of phenotypic HRV, and (3) genetically predicted HRV with incident mortality.

We performed a GWAS on 46,075 individuals of European ancestry participating in the UK Biobank to increase our understanding of genes that influence HRV. We additionally investigated the longitudinal association of HRV at baseline ($N = 54,312$) and incident mortality during a median follow-up period of seven years. We constructed genetic risk scores (GRSs) for the remaining independent sub-sample of the UK Biobank ($N = 412,891$) with genotype data, but not included in the GWAS, to further our understanding of molecular mechanisms linking reduced HRV with increased risk of all-cause and cardiovascular mortality.

## Results

**General characteristics**. Supplementary Data 1 presents the general characteristics of participants included in this study. A total of 46,075, 54,312, and 412,891 participants with a mean age of 57 years were included for the GWAS, phenotypic, and GRS association analyses, respectively. Overall, 1118 (2.4%) participants in the GWAS sample, 1548 (2.9%) in the phenotypic sample, and 17,503 (4.2%) in the GRS sample died of any cause

during the follow-up periods (median: 6.99, 6.99, and 8.14 years, respectively).

The GWAS on the four natural log-transformed HRV traits (RMSSD, RMSSDc, SDNN, and SDNNc) identified 23 genetic variants (17 independent) at a genome-wide significance level ($P < 5 \times 10^{-8}$). In our study, a locus was defined 2.5 Mb distance up- and downward of the sentinel SNP. Based on this, there were 16 loci. Notably, eight of these loci (*RNF220, GNB4, LINCR_0002, KLHL3/HNRNPA0, CHRM2, KCNJ5, MED13L,* and *C160rf72*) have not been previously reported for HRV (Table 1 and Fig. 1) and were not in LD ($r^2 < 0.005$) with or within 2.5 Mb of a previously reported locus. Six previously reported loci (*RGS6, PPIL1, SYT10, GNG11, LINC00477,* and *NDUFA11*) were associated with all HRV traits (Fig. 2). In a lookup of the 23 association signals in the summary statistics of a previously published HRV GWAS[9], only 16 were present and for the remaining seven genetic variants we used their proxies ($r^2 > 0.7$). Twenty-one of the 23 variants replicated with a one-sided $P < 0.05$. Two genetic variants (one indel) did not reach nominal significance (one-sided $P < 0.05$). A further six variants (one indel) did not reach the more conservative one-sided Bonferroni-corrected threshold of 0.00294 (0.05/17) (Supplementary Data 14). The quantile–quantile and regional association plots are given in Supplementary Figs. 2, 3–19, respectively. A detailed description of what is known on biological function of genes reported in this manuscript is given in Supplementary Note 1. As sensitivity analyses, we reran the GWAS after excluding individuals who died within 1 or 2 years after the ECG recording. The result shows that the effect sizes of the significant variants in the original analyses remain virtually the same, indicating that our results were not biased by prevalent diseases (Supplementary Data 13).

From the newly reported genes, the lead variant (rs71784944: effect allele frequency 0.226) of the *CHRM2* locus on chromosome 7 was an indel and associated only with RMSSD. The frequency of this variant for non-Finnish Europeans in The Genome Aggregation Database (gnomAD). (https://gnomad.broadinstitute.org/) is 0.217. There were two lead SNPs annotated to the *KCNJ5* gene, rs7609764 which was associated with both SDNN and RMSSD and rs7102584 associated with only RMSSD. The frequency of rs7102584 is relatively low both in UKB (0.017) and in gnomAD non-Finnish Europeans (0.012) (Table 1).

Of the 17 previously reported SNP associations (11 independent) for HRV by Nolte et al.[9], we replicated 16 ($P < 0.05/11 = 0.00455$) confirming all previously reported loci (Supplementary Data 7).

Using LDSC, we estimated SNP-based heritability to be 9.45% for RMSSD and 7.86% for SDNN. The genetic correlation between RMSSD and SDNN was 98.4% (Supplementary Data 8). We also estimated genetic correlations ($r_g$) of HRV related with cardiac, metabolic and neuropsychiatric traits using GWAS summary statistics. From the 11 traits tested (Supplementary Data 9), we found evidence of shared genetic effects of RMSSD with diastolic blood pressure ($r_g = -0.25$, $P = 1.57 \times 10^{-9}$), systolic blood pressure ($r_g = -0.14$, $P = 0.0013$), heart rate ($r_g = -0.74$, $P = 1.48 \times 10^{-20}$), heart failure ($r_g = -0.23$, $P = 9.00 \times 10^{-4}$), coronary artery disease ($r_g = -0.16$, $P = 3.00 \times 10^{-4}$), and type 2 diabetes ($r_g = -0.48$, $P = 0.035$).

The in silico sequencing analyses of the 23 genetic variants resulted in 1433, 452, and 39 SNPs that were in moderate ($r^2 > 0.5$), high ($r^2 > 0.8$), and perfect LD ($r^2 = 1$), respectively. Annotation of these linked SNPs using the ANNOVAR software detected a missense lead SNP (rs7102584) at *KCNJ5*, a novel locus (Supplementary Data 6). This SNP was characterized as tolerated, with a SIFT score of 1, indicating no possible damaging effect. We also identified a missense variant in the *NDUFA11* locus

**Table 1 Genome-wide significant variants in 16 loci associated with HRV traits.**

| Locus | SNP | CHR | BP (hg19) | Closest gene | Trait | EA/OA | EAF | Beta | SE | P value |
|---|---|---|---|---|---|---|---|---|---|---|
| 1 | rs156653 | 1 | 45003255 | **RNF220** | RMSSD | T/C | 0.277 | −0.023 | 0.004 | 4.30E-08 |
| 2 | rs7612445 | 3 | 179172979 | **GNB4** | RMSSD | G/T | 0.803 | −0.029 | 0.005 | 5.70E-10 |
| | | | | | RMSSDc | | | −0.023 | 0.004 | 2.00E-08 |
| 3 | rs1083698 | 3 | 191413314 | **LINCR-0002** | RMSSD | C/T | 0.583 | −0.022 | 0.004 | 5.60E-09 |
| | | | | | RMSSDc | | | −0.019 | 0.003 | 2.60E-08 |
| 4 | rs56210945[a] | 5 | 137091161 | **HNRNPA0** | RMSSD | C/A | 0.821 | −0.030 | 0.005 | 1.20E-09 |
| | rs2905583[a] | 5 | 137003769 | **KLHL3** | RMSSDc | G/T | 0.818 | −0.025 | 0.004 | 4.40E-09 |
| 5 | rs236349 | 6 | 36820565 | PPIL1 | RMSSD | G/A | 0.658 | −0.032 | 0.004 | 6.30E-16 |
| | | | | | RMSSDc | | | −0.029 | 0.003 | 4.50E-17 |
| | | | | | SDNN | | | −0.033 | 0.004 | 9.20E-20 |
| | | | | | SDNNc | | | −0.031 | 0.003 | 4.00E-19 |
| 6 | rs71784944 | 7 | 136603315 | **CHRM2** | RMSSD | GT/G | 0.226 | −0.025 | 0.005 | 2.90E-08 |
| 7 | rs756675674[b] | 7 | 93539158 | GNG11 | SDNNc | GA/G | 0.459 | −0.022 | 0.004 | 4.10E-10 |
| | rs180251[b] | 7 | 93546256 | | RMSSD | C/T | 0.351 | −0.032 | 0.004 | 1.30E-16 |
| | rs180244[b] | 7 | 93549363 | | RMSSDc | G/C | 0.349 | −0.027 | 0.003 | 3.00E-15 |
| | | | | | SDNN | | | −0.027 | 0.004 | 5.60E-13 |
| 8 | rs76097649[c] | 11 | 128764570 | **KCNJ5** | RMSSD | G/A | 0.914 | −0.052 | 0.007 | 5.20E-15 |
| | | | | | RMSSDc | | | −0.044 | 0.006 | 6.20E-14 |
| | | | | | SDNN | | | −0.037 | 0.006 | 3.40E-09 |
| 9 | rs7102584[c] | 11 | 128782012 | | RMSSD | C/G | 0.017 | −0.087 | 0.015 | 5.30E-09 |
| | | | | | RMSSDc | | | −0.071 | 0.013 | 4.60E-08 |
| 10 | rs802122 | 12 | 116247694 | **MED13L** | RMSSD | G/A | 0.632 | −0.024 | 0.004 | 4.10E-10 |
| | | | | | RMSSDc | | | −0.020 | 0.003 | 5.20E-09 |
| 11 | rs4963772 | 12 | 24758480 | LINC00477 | RMSSD | G/A | 0.849 | −0.065 | 0.005 | 3.00E-35 |
| | | | | | RMSSDc | | | −0.054 | 0.005 | 7.60E-32 |
| | | | | | SDNN | | | −0.054 | 0.005 | 8.40E-29 |
| | | | | | SDNNc | | | −0.043 | 0.005 | 1.50E-21 |
| 12 | rs6488162 | 12 | 33593127 | SYT10 | RMSSD | T/C | 0.419 | −0.041 | 0.004 | 1.70E-26 |
| | | | | | RMSSDc | | | −0.034 | 0.003 | 1.50E-23 |
| | | | | | SDNN | | | −0.030 | 0.004 | 3.00E-17 |
| | | | | | SDNNc | | | −0.023 | 0.003 | 6.50E-12 |
| 13 | rs1906263[d] | 12 | 38653362 | ALG10B | RMSSD | C/T | 0.427 | −0.025 | 0.004 | 9.10E-11 |
| | | | | | RMSSDc | | | −0.021 | 0.003 | 7.40E-10 |
| | rs35861884[d] | 12 | 38791366 | | RMSSD | A/G | 0.439 | −0.022 | 0.004 | 3.30E-08 |
| 14 | rs17180489 | 14 | 72885471 | RGS6 | RMSSD | G/C | 0.861 | −0.043 | 0.005 | 2.40E-15 |
| | | | | | RMSSDc | | | −0.034 | 0.005 | 9.00E-13 |
| | | | | | SDNN | | | −0.041 | 0.005 | 4.40E-16 |
| | | | | | SDNNc | | | −0.032 | 0.005 | 8.80E-12 |
| 15 | rs1979409 | 15 | 73465477 | NEO1 | RMSSD | A/G | 0.455 | −0.023 | 0.004 | 1.20E-09 |
| | 16:9290009_TA_T | 16 | 9290009 | **C16orf72** | SDNN | T/TA | 0.201 | −0.025 | 0.004 | 1.80E-08 |
| | | | | | SDNNc | | | −0.023 | 0.004 | 3.50E-08 |
| 16 | rs201334918[e] | 19 | 5881576 | NDUFA11 | RMSSD | TA/T | 0.084 | −0.092 | 0.007 | 3.90E-42 |
| | rs35952442[e] | 19 | 5889071 | | RMSSDc | T/C | 0.083 | −0.093 | 0.006 | 6.10E-56 |
| | rs12974991[e] | 19 | 5894584 | | SDNN | A/G | 0.082 | −0.058 | 0.006 | 3.50E-20 |
| | | | | | SDNNc | | | −0.061 | 0.006 | 1.10E-24 |

RMSSD root mean square of successive differences, RMSSDc corrected root mean square of successive differences, SDNN SD of normal-to-normal intervals, SDNNc corrected SD of normal-to-normal intervals, EA effect allele, OA other allele, EAF effect allele frequency, CHR chromosome, BP base pair position based on build 37 (hg19), SE standard error.
Note: Only SNPs that were independently associated (that is, lead SNPs) to HRV traits are shown. At some loci, lead SNPs were the same for the different traits, at other loci there were different (dependent) lead SNPs for the different traits. Loci that were not previously reported at the time of analysis are reported in bold.
[a] $r^2 = 0.899$ between rs56210945 and rs2905583.
[b] $r^2 = 0.707$ between rs756675674 and rs180251, $r^2 = 0.704$ between rs756675674 and rs180244, and $r^2 = 0.988$ between rs180251 and rs180244.
[c] $r^2 = 0.001$ between rs76097649 and rs7102584.
[d] $r^2 = 0.452$ between rs1906263 and rs35861884.
[e] $r^2 = 0.988$ between rs201334918 and rs35952442, $r^2 = 0.984$ between rs201334918 vs rs12974991, and $r^2 = 0.996$ between rs35952442 and rs1297499.

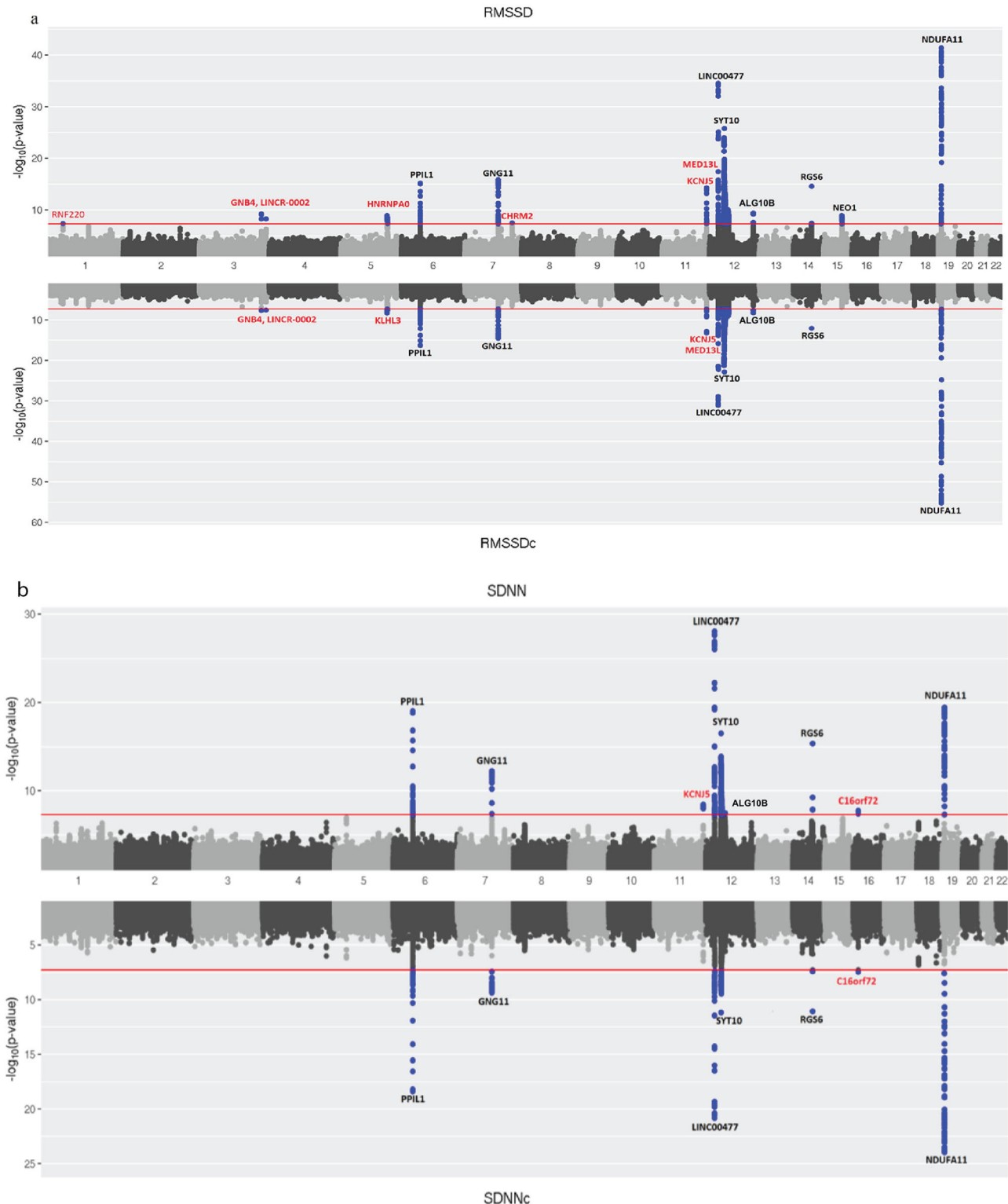

**Fig. 1 Mirrored Manhattan plots of the GWAS of HRV traits. a** RMSSD and RMSSDc and **b** SDNN and SDNNc. The red horizontal line represents the genome-wide significance threshold. Genes closest to the independent lead SNPs are indicated for the loci that were genome-wide significantly associated with the trait. Novel loci are highlighted in red. RMSSD root mean square of successive differences, RMSSDc hear rate-corrected root mean square of successive differences, SDNN SD of normal-to-normal intervals, SDNNc heart rate-corrected SD of normal-to-normal intervals.

(rs12980262; in perfect LD [$r^2 = 1$] with sentinel variants rs35952442 and rs12974991), which was previously associated with HRV. This SNP had a SIFT score of 0.01 and a PolyPhen score of 0.753, suggesting deleterious effects and likely functional consequences.

In silico pleiotropy analysis indicated that the lead SNPs were previously reported to be associated with HRV (rs236349, rs4963772, and rs12974991), (resting) heart rate (rs7612445 and rs17180489), heart rate increase in response to exercise, and heart rate response to recovery post exercise (rs236349, rs4963772,

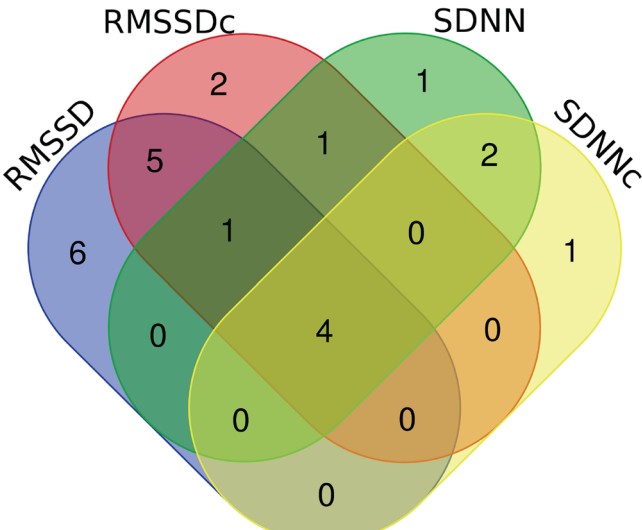

**Fig. 2 Venn-diagram of lead SNPs shared by the HRV traits.** RMSSD root mean square of successive differences, RMSSDc heart rate-corrected root mean square of successive differences, SDNN SD of normal-to-normal intervals, SDNNc heart rate-corrected SD of normal-to-normal intervals.

rs6488162, rs17180489, and rs12974991), atrial fibrillation (rs7612445 and rs76097649), pulse pressure (rs71784944 and rs4963772) and familial hyperaldosteronism (rs7102584). In addition, we found that SNPs in high LD ($r^2 > 0.8$) with eight and 11 of the lead SNPs were previously associated with HRV and heart rate, respectively. Six of our 23 sentinel SNPs had not been identified in any previous GWASs (Supplementary Data 5).

**Associations of HRV traits with all-cause and cardiovascular mortality**. In Cox-regression analyses using the phenotypic HRV data in 54,312 individuals, we found that lower HRV values were associated with more considerable hazards of all-cause mortality. For each unit of natural log-transformed RMSSD and SDNN decrease, individuals had 19% (HR: 1.19, 95% CI 1.10–1.28; $P = 1.48 \times 10^{-05}$) and 32% (HR: 1.32, 95% CI 1.23–1.43; $P = 9.30 \times 10^{-12}$) higher risk of dying from any cause during the follow-up period, respectively (Fig. 3 and Supplementary Data 2). These associations were similar for the heart rate-corrected HRV measures. Only SDNN was significantly associated with cardiovascular mortality (HR: 1.28, 95% CI: 1.06–1.52; $P = 6.86 \times 10^{-3}$), but this association was no longer significant when SDNN was corrected for the mean IBI (HR: 1.19, 95% CI: 0.98–1.45; $P = 7.21 \times 10^{-2}$) (Fig. 3). We performed Kaplan–Meier analyses to compare the risk of death for quartiles of HRV traits (considering the 4th quartile with the highest HRV values as a reference) (Figs. 4 and 5 and Supplementary Data 3). Participants in the 1st (lowest) quartile at baseline for RMSSD and SDNN had 1.31 (95% CI: 1.15–1.51; $P = 9.29 \times 10^{-5}$) and 1.46 (95% CI: 1.27–1.68; $P = 1.16 \times 10^{-7}$) times higher risk of death from any cause compared to participants in the highest quartile. Our sensitivity analyses show that even after excluding individuals who died within 1 and 2 years after ECG measurement, lower HRV values were associated with a significant hazard of all-cause mortality (Supplementary Data 12). We did not find significant differences in the risk of death from cardiovascular causes for participants with the lowest and highest quartile of RMSSD and SDNN (HR: 1.34, 95% CI: 0.98–1.82; $P = 0.062$ and HR: 1.15, 95% CI: 0.85–1.57; $P = 0.360$). For our sub-analyses for death from cancer causes, like the all-cause mortality, participants in the lowest quartile for RMSSD and SDNN had 1.43 (95% CI: 1.16–1.76; $P = 8.05 \times 10^{-4}$) and 1.64 (95% CI: 1.32–2.05;

$P = 8.94 \times 10^{-6}$) times higher risk of death from cancer compared to participants in the highest quartile (Supplementary Data 11).

**Associations of genetically determined HRV traits with all-cause, cancer and cardiovascular mortality**. We constructed weighted GRSs of HRV-decreasing alleles to evaluate the associations between genetically determined HRV with all-cause and cardiovascular mortality. None of them were significantly associated with all-cause or cardiovascular mortality (Supplementary Data 4 and 5 and Supplementary Figs. 20 and 21). Similarly, genetically determined HRV was not significantly associated with cancer mortality (Supplementary Data 4 and 5 and Supplementary Fig. 22).

**Discussion**
In this study, a GWAS of 46,075 individuals of European ancestry, we identified 23 genome-wide significant genetic variants in 16 loci associated with HRV of which 8 of these loci have not been reported previously. In phenotypic analyses investigating the association of HRV and mortality, we found that lower HRV values at baseline were associated with a higher risk of all-cause and cancer mortality but not consistently with cardiovascular mortality. Furthermore, the genetic variants associated with low HRV combined in a GRS were not associated with mortality.

Our study confirms previously reported genetic loci (*RGS6*, *NEO1*, *PPIL1*, *SYT10*, *GNG11*, *LINC00477*, *NDUFA11*) in a meta-analysis of GWAS for HRV[9]. Many of the HRV loci harbor causal candidates that are involved in cardiac development (*RNF220*, *NDUFA11*), neural development (*SYT10*), electrophysiological processes (*RGS6*, *GNG11*), and muscarinic cholinergic receptor function (*CHRM2*). The *NDUFA11* gene harbors a likely deleterious variant, which may have functional consequences. *NDUFA11* is an accessory subunit of the mitochondrial membrane respiratory chain NADH dehydrogenase complex I. In humans, a splice-site mutation in this gene is known to cause mitochondrial complex I deficiency. A recent study on mice[11] has shown that downregulation of *NDUFA11* reduced ATP production and increased mitochondria reactive oxygen species production in cardiac mitochondria. A detailed description of the known biological function of genes reported in this manuscript is given in Supplementary Note 1.

Three of the loci that have not been reported previously, *CHRM2*, *GNB4*, and *KCNJ5*, perfectly fit in previously depicted pathways of cardiac vagal effects on the sinoatrial node and the potential roles of HRV loci[9].

Interestingly, we found, *CHRM2*, a previously investigated candidate gene with functions in the acetylcholine pathway[7], to be associated with HRV. *CHRM2* encodes the muscarinic acetylcholine receptor M2, which is located both pre-and post-synaptically. This receptor is known for its negative chronotropic and inotropic effects after binding with acetylcholine released by postganglionic parasympathetic nerves. Posokhova et al.[12] showed that the muscarinic receptor affects HRV by acetylcholine-dependent inward-rectifier potassium current (IKACh) in mice. *CHRM2* is found in the sinoatrial node of the heart among the different tissues involving in both heart rate and cardiac function. This gene has been previously reported to be associated with resting heart rate[13,14], heart rate response to post exercise recovery[15,16], and pulse pressure measurement[17,18].

Heterotrimeric guanine nucleotide-binding proteins (G proteins) are involved as a modulator or transducer in various transmembrane signaling systems, including vagal-muscarinergic signaling underlying HRV, with an alpha, a beta, and a gamma subunit. *GNB4* gene encodes beta subunits which are important regulators of alpha subunits, as well as of certain signal

| HRV traits | Hazard ratio (95%CI) | P-value |
|---|---|---|
| **All-cause mortality** | | |
| lnRMSSD | 1.19 (1.10-1.28) | 1.48E-05 |
| lnRMSSDc | 1.12 (1.04-1.23) | 6.16E-03 |
| lnSDNN | 1.32 (1.23-1.43) | 9.30E-12 |
| lnSDNNc | 1.27 (1.15-1.34) | 3.50E-07 |
| **Cardiovascular mortality** | | |
| lnRMSSD | 1.14 (0.96-1.33) | 1.44E-01 |
| lnRMSSDc | 1.05 (0.88-1.28) | 5.64E-01 |
| lnSDNN | 1.28 (1.06-1.52) | 6.86E-03 |
| lnSDNNc | 1.19 (0.98-1.45) | 7.21E-02 |

**Fig. 3 Phenotypic associations of HRV traits with all-cause and cardiovascular mortality.** lnRMSSD log-tranformed root mean square of successive differences, lnRMSSDc log-tranformed corrected root mean square of successive differences, lnSDNN log-transformed SD of normal-to-normal intervals, lnSDNNc log-transformed corrected SD of normal-to-normal intervals, CI confidence interval.

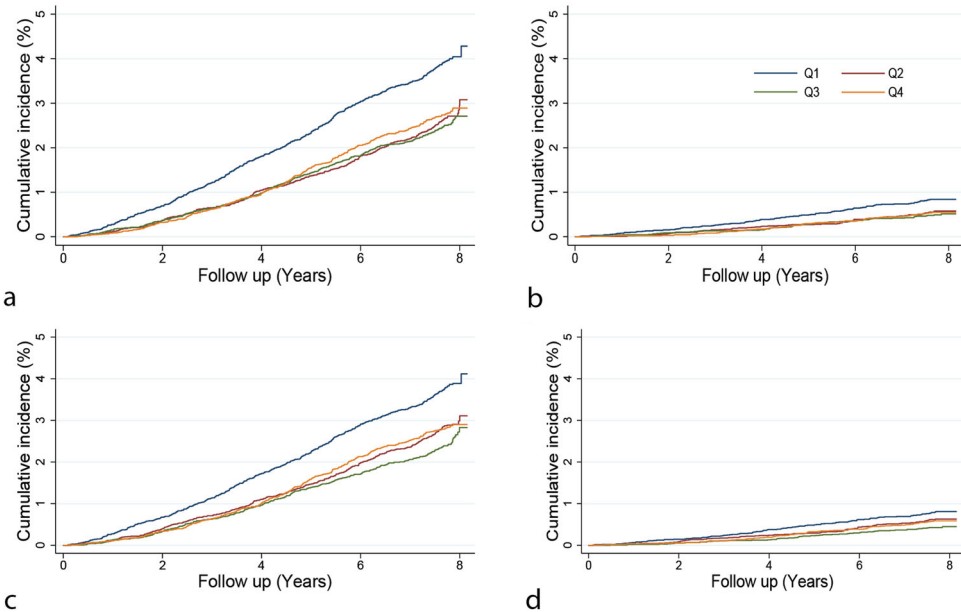

**Fig. 4 Kaplan–Meier curves for risk of mortality among participants in quartiles of HRV traits. a** RMSSD and all-cause mortality; **b** RMSSD and cardiovascular mortality; **c** RMSSDc and all-cause mortality; **d** RMSSDc and cardiovascular mortality. RMSSD root mean square of successive differences, RMSSDc corrected root mean square of successive differences.

transduction receptors and effectors like the G protein-gated inwardly rectifying potassium (GIRK) channel. A genome-wide identification of Expression Quantitative Trait Loci (eQTLs) in cardiac tissue by Koopmann et al.[19] reported that *GNB4* is a left ventricular myocardium eQTL. In addition, *GNB4* has been previously reported for resting heart rate[13].

*KCNJ5* gene, also known as *GIRK4*, is involved in the transmembrane transfer of potassium ions by a voltage-gated channel through the plasma membrane of atrial cardiomyocytes, contributing to the repolarization phase of the action potential. The lead SNP at the *KCNJ5* locus, rs7102584, is a missense variant, which is not indicated as deleterious. *KCNJ5* has been previously reported for resting heart rate[14], atrial fibrillation[20,21], and diastolic blood pressure[18].

One of the major findings of the present study was that individuals with lower HRV values at inclusion had a higher risk of dying from any cause during the follow-up period. This finding is consistent across the different HRV traits studied and remains significant even after adjusting for heart rate. We find that associations between uncorrected HRV and morbidities/mortalities remain after correction for heart rate in the corrected HRV measures showing these associations are largely driven by HRV (mainly reflecting parasympathetic influences on the heart) rather than just heart rate (mainly reflecting the influence of the sympathetic nervous system). Consistent with our results, the Atherosclerosis Risk In Communities (ARIC) Study also reported that low HRV was predictive of increased mortality rates[22]. In addition to our study which was based on the general population, several previous studies among patient populations showed similar results with reduced HRV as a significant predictor of all-cause mortality. For instance, Arildsen et al. reported baseline HRV could be used to predict all-cause mortality after five years of follow-up in a diabetic population[23]. In addition, the Chronic Renal Insufficiency Cohort (CRIC) study reported HRV, as

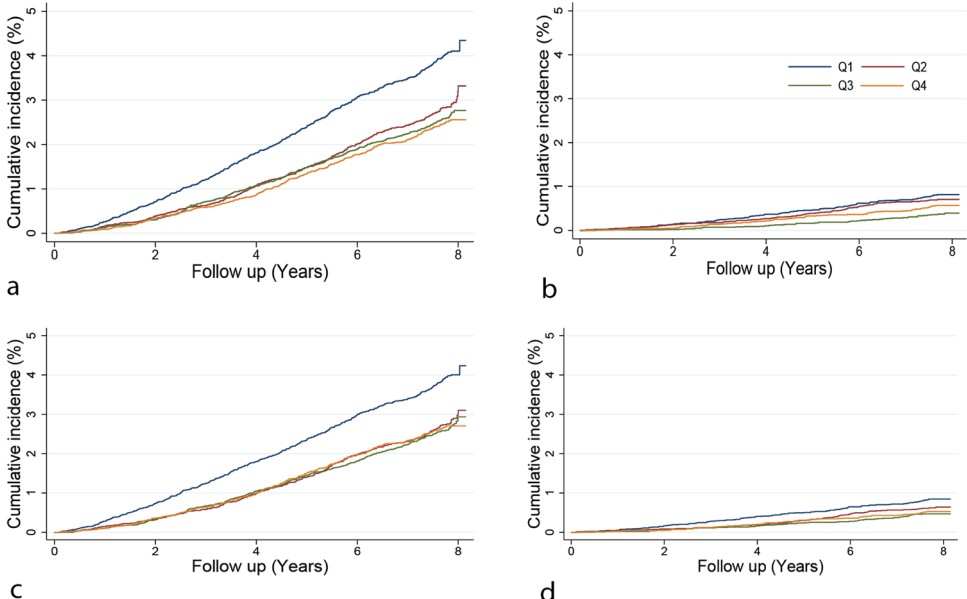

**Fig. 5 Kaplan–Meier curves for risk of mortality among participants in quartiles of HRV traits. a** SDNN and all-cause mortality; **b** SDNN and cardiovascular mortality; **c** SDNNc and all-cause mortality; **d** SDNNc and cardiovascular mortality. SDNN SD of normal-to-normal intervals, SDNNc corrected SD of normal-to-normal intervals.

measured by RMSSD, was associated with all-cause mortality. In this report, both low and high RMSSD was associated with increased risk for all-cause mortality. The latter might be due to sinoatrial node dysfunction[24] rather than autonomic dysfunction. Similarly, our results show individuals with lower HRV values had a higher risk of dying from cancer during the follow-up consistent with a systematic review by Kloter et al.[25] that concluded that individuals with higher HRV and advanced coping mechanisms seem to have a better prognosis in cancer progression. The possible explanations might include a lower HRV is associated with tumor growth through inflammation, oxidative stress, and sympathetic nerve activation.

Likewise, participants with lower SDNN values showed a higher risk to develop cardiovascular mortality. In agreement with this, a meta-analysis by Hillebrand et al.[26] reported that a low SDNN was associated with an increased risk of cardiovascular death. The effect, however, disappeared when heart rate was added as an additional predictor, possibly because high heart rate and low HRV reflect lower cardiac vagal drive, but the alternative explanation is that higher heart rate is the primary risk factor for cardiovascular death, while simultaneously reducing HRV[27].

In this study, we also showed that genetic variants associated with low HRV were not associated with a change in all-cause or cardiovascular mortality risk. Nonetheless, the prospective association in our phenotypic analyses and prior studies clearly show a strong relationship between low HRV and a higher risk of all-cause mortality. This leaves two likely possibilities: either the genetic instruments used in our analyses were not strong enough, or the association between HRV and all-cause mortality is not causal, but reflects confounding environmental effects (e.g., chronic stress) that can independently reduce HRV as well as increase all-cause mortality. The latter is more likely to be accurate as the large sample size in our study was well-powered to detect a potential genetic association for our GRSs. Power calculations using mRnd[28] also showed that the power was optimal (100%) for both RMSSD and SDNN for both all-cause and cardiovascular mortality based on realistic effect size assumptions (Supplementary Data 10). Thus, weak instrument bias for our GRSs was not a concern. Further research is therefore needed to

explore the driving mechanisms behind the phenotypic association found in both our and previous analyses.

We conducted the largest discovery GWAS to date for HRV in a European sample. In addition, although the associations between HRV and mortality have been studied previously, our study provide long-term follow-up data in a very large sample. Nonetheless, our study has some limitations. First, some of the genome-wide significant genetic variants could not be found in the replication dataset. However, we used their proxies ($r^2 > 0.7$) to evaluate the replication, but these are obviously suboptimal. Second, our analyses were restricted to European ancestry, and this may reduce the generalizability of the results to other ethnicities. A third limitation is that the ultrashort nature of the ECG recording in the UKB limited us to assess only the time-domain HRV indices. This was because at least a one-minute ECG recording is recommended to estimate frequency-domain values[29]. We speculate that future inclusion of diverse ethnic populations and HRV measures may give additional insights into the genetic architecture of HRV and will further contribute to test causal hypotheses on the association of HRV and health outcomes. Finally, the limited number of cardiovascular events in our study might have resulted in an underestimation of the (genetic) association between HRV and cardiovascular mortality.

In conclusion, this study identified new variations that are genome-wide significant for HRV in genes involved in cardiac development and electrophysiology function and it underscores that HRV is a complex trait controlled by multiple genes. To the best of our knowledge, the findings provide novel biological insights into the mechanisms underlying HRV. The present study clearly demonstrates that reduced HRV had a strong association with a higher risk of all-cause mortality in an adult population. These results also emphasize the role of the cardiac autonomic nervous system, as indexed by HRV, in predicting mortality. No evidence was found for an association between genetic predictors of HRV with mortality.

## Methods

**Study setting and population.** This study uses data from the UK Biobank, a prospective cohort study of 503,325 individuals from the United Kingdom with an age range of 40–69 who were

recruited between 2006 and 2010. The Northwest Multi-Centre Research Ethics Committee approved the UK Biobank study, and all participants provided written informed consent to participate in the study. A detailed description of the methods used by UK Biobank is available elsewhere[30].

During the baseline and follow-up visit, a sub-sample of the UK Biobank participants (Supplementary Fig. 1) underwent a cardio assessment using a stationary bicycle with a 4-lead electrocardiogram (ECG) device to record ECGs at pre-test (15 s), during activity (6 min) and recovery (1 min). We used the pre-test recording (i.e., resting) ECG assessments at first visit with a full disclosure to use the recordings ($n = 63,924$) to calculate HRV indices for this paper (Supplementary Fig. 1). Participants were asked to sit still on a chair, relaxed and quiet. They were allowed to remove their jacket or loosen clothing before placing the ECG electrodes. The cardio assessment involved a three lead (lead I, II, and III) ECG recording (AM-USB 6.5, Cardiosoft v6.51) using four electrodes placed on the antecubital fossa (right and left) and wrist (right and left). Participants fitted with a pacemaker did not complete the resting ECG procedure. We used in-house software, described elsewhere[31], to detect R-peaks and calculate HRV indices. HRV could not be calculated in 4260 participants due to excessive noise and ectopic (non-sinus node) beats (Supplementary Fig. 1).

We calculated the root mean square of successive differences (RMSSD) and the standard deviation of normal-to-normal intervals (SDNN) as indices of HRV. Heart rate has a strong relationship with HRV, which includes a mathematical dependency of the variance on the mean inter-beat interval (IBI) that is difficult to separate from the joint vagal effects on IBI and HRV[32]. Van Roon et al. reintroduced an approach recommended by Akselrod et al.[33] to correct HRV for its dependency on the mean IBI of consecutive R-peaks using coefficients of variation[34]. The coefficient of variation detects the amount of IBI variability relative to the mean IBI of each participant. We applied this method to additionally calculate HRV values that were corrected for the influence of mean IBI, that is the RMSSDc and SDNNc.

Non-Europeans ($n = 5352$) were excluded to get a homogeneous population of 54,312 participants for the phenotypic association analysis between HRV and mortality. European ancestry was defined based on self-reported ethnicity by participants. For the GWAS sample, participants were also excluded if they had missing or poor-quality genotype data, missing covariates, a diagnosis of angina, myocardial infarction, heart failure, or used anti-depressant medication, digoxin, atropine, or acetylcholinesterase (Supplementary Fig. 1).

**Mortality**. All-cause mortality included all deaths that occurred after inclusion and before Jan 31, 2018, for participants from England or Wales and before Nov 30, 2016, for participants from Scotland. Data on deceased UK Biobank participants have been received from the National Health Service (NHS) Digital for participants in England and Wales and the NHS Central Register (NHSCR), part of the National Records of Scotland, for participants in Scotland. Cardiovascular and cancer mortality was defined based on the data UK Biobank received from the death registry, including the date of death and the primary and contributory causes of death, coded using the ICD-10 system. Details of mortality data linkage to death registries are available on http://biobank.ctsu.ox.ac.uk/crystal/crystal/docs/DeathLinkage.pdf.

**Covariates**. Age and sex were included as covariates for the GWAS and phenotypic analyses. In addition, diseases that are known to influence HRV were included as covariates for the phenotypic analysis. Information on these diseases, such as

hypertension, type 2 diabetes, angina, myocardial infarction, and heart failure, was obtained from self-reports (touch screen questionnaire and verbal interview) and/or captured through the Hospital Episode Statistics records[35] using the following International Classification of Diseases (ICD) codes: I10–I15 for hypertension; E10–E14 for diabetes; I21–I22 for myocardial infarction and I42, I150 for heart failure.

**Genotyping and imputation**. Genotyping, quality control, and imputation of the UK Biobank have been described elsewhere[36]. In brief, 488,377 participants have been genotyped using the custom UK Biobank Axiom array ($n = 438,427$) or UK Biobank Lung Exome Variant Evaluation (UK BiLEVE) Axiom array ($n = 49,950$). These arrays have >95% common variant content. In addition, Haplotype Reference Consortium, 1000 Genomes Project phase 3 release, and UK10K reference panels were used for imputation resulting in 93,095,623 autosomal SNPs, short insertion/deletions (indels), and large structural variants in 487,442 individuals and an additional 3,963,705 markers on the X chromosome.

### Statistics and reproducibility
*Association of HRV phenotypes with mortality.* We assessed the association between HRV phenotypes (RMSSD, RMSSDc, SDNN, SDNNc) and time to death using Cox-regression analyses adjusted for age, age², sex, and diseases (angina, myocardial infarction, heart failure, type 2 diabetes, and hypertension). We also constructed Kaplan–Meier curves for quartiles of HRV traits (considering the 4th quartile as a reference). Similar analyses were performed for cardiovascular mortality.

*Genome-wide association study.* We performed GWAS analyses for four natural log-transformed HRV traits (RMSSD, RMSSDc, SDNN, and SDNNc) in 46,075 European participants using BOLT-LMM[37], which employs a conjugate gradient-based iterative framework for fast mixed-model computations to account for population structure and relatedness accurately. Additive genetic effects were assumed. Analyses were adjusted for age at inclusion, age², sex, genotyping chip (UK Biobank Axiom or UK BiLEVE), the first 30 genetic principal components provided by the UK Biobank, and the Townsend deprivation index as a proxy for socioeconomic status. Genetic variants with minor allele frequencies <0.005 or INFO scores <0.05 were excluded from the analyses. We used the GWASinspector package[38] to perform quality control of GWAS results. SNPs that passed the genome-wide significance threshold of $P < 5 \times 10^{-8}$ were clumped together using the clumping procedure in PLINK 1.9, with linkage disequilibrium (LD) $r^2 > 0.005$ and 2.5 Mb distance. The genetic variant with the smallest $P$ value within a locus was designated the sentinel genetic variant.

A lookup of the identified genome-wide significant genetic variants was performed in meta-analyzed GWAS data of 20 cohorts of European ancestry in up to 28,700 participants[9]. This dataset used HapMap2 as a reference panel and was therefore imputed using a 1000 Genome reference panel before the lookup using FiZi[39]. An association was considered replicated if the direction of effect was consistent and the one-sided replication $P$ value was <0.05. We also evaluated replication for a more conservative Bonferroni-corrected one-sided threshold ($P < 0.00294$) corrected for the total number of 17 independent SNPs (0.05/17).

*Heritability and genetic correlation estimation.* LD Score Regression (LDSC)[40] was used to estimate the SNP-based heritability explained by common variants, and bivariate genetic correlations

of HRV with a range of cardiac, metabolic, and neuropsychiatric outcomes: blood pressure, heart rate, heart failure, atrial fibrillation, PR interval, coronary artery disease, type 2 diabetes, depressive symptoms, Alzheimer's disease, and Parkinson's disease.

*Genetic risk score.* We constructed weighted GRSs of HRV-decreasing alleles to evaluate the associations between genetically determined lower HRV with all-cause and cardiovascular mortality. These weighted GRSs were constructed for each HRV phenotype by multiplying the effect sizes of all associated genetic variants (see Table 1) with their allele dosages (range 0–2) and then summing all variants together. All European UK Biobank participants who were not part of the discovery GWAS, but did have genotype information ($n = 412{,}891$, Supplementary Fig. 1), were included to assess the association of the GRS with all-cause and cardiovascular mortality. We performed Cox-regression analyses adjusted for age at inclusion, age$^2$, sex, body mass index (BMI), genotyping chip, and the first ten principal components provided by the UK Biobank. We also constructed Kaplan–Meier curves for quartiles of the GRS (considering the 4th quartile as a reference). To ensure the strength of GRSs as instrumental variables (IVs), we generated an *F*-statistic for each outcome. We used variance in lnRMSSD and lnSDNN explained by each set of HRV SNPs (conservatively estimated at 2.4% and 1.4%, respectively, based on ref. [9]) to calculate the *F*-statistic using the formula $F\text{-statistic} = [R^2 \times (n - 1 - K)]/[(1 - R^2) \times K]$, where $R^2$ represents the proportion of variability in RMSSD and SDNN level that is explained by the GRS, $n$ represents sample size, and $K$ represents the number of IVs included in model (i.e., for this study $K = 1$)[41]. As a rule of thumb, an *F*-value above ten indicates that a causal estimate is unlikely to be biased due to weak instruments. Based on these calculations, the F-statistics for RMSSD and SDNN were 237 and 80, respectively, confirming the strength of our GRS IVs.

*Functional variants and candidate causal genes.* In silico sequencing of the significant genetic loci from the GWAS defined by 1 Mb regions on either side of the independent top SNPs was done using the European data of the 1000 Genomes project phase 3 as a reference panel. A cut-off value for LD was set at $r^2 > 0.50$ to identify potentially functional missense variants in LD with the sentinel SNP. Annotation of the significant genetic variants together with their linked SNPs was carried out using the ANNOVAR software[42]. We used the Sorting Intolerant From Tolerant (SIFT)[43] and Polymorphism Phenotyping (PolyPhen)[44] scoring tools to predict the possible damaging effects of nonsynonymous (ns) SNPs on protein structure and function.

*Pleiotropy analyses.* In silico pleiotropic effects of HRV-associated genetic variants and their linked variants with $r^2 > 0.50$ with other traits or diseases reported in previous GWAS studies available in the National Human Genome Research Institute (NHGRI)[45] (accessed on October 1, 2020) were assessed using the GWAS Catalog database (https://www.ebi.ac.uk/gwas/).

*Sensitivity analyses.* To minimize the influence of bias introduced by prevalent diseases, we excluded participants who died soon after the ECG recording. We repeated the phenotypic and GWAS analyses after excluding individuals who died within 1 and 2 years after the ECG measurements separately. In addition to all-cause mortality and cardiovascular mortality, we assessed the association between HRV phenotypes and GRSs with cancer mortality since cancer was one of the most common causes of death.

**Reporting summary.** Further information on research design is available in the Nature Portfolio Reporting Summary linked to this article.

## Data availability
Access of raw data is restricted to protect the privacy of participants, but it can be requested via UK Biobank (www.ukbiobank.ac.uk). The GWAS summary statistics are publicly available in the NHGRI-EBI GWAS catalog at https://www.ebi.ac.uk/gwas/ (accession IDs: GCST90281263, GCST90281264, GCST90281265, and GCST90281266). The source data used to plot Fig. 3 are provided in Supplementary Data 2. The source data used to generate Figs. 4 and 5 are provided in Supplementary Data 3.

## Code availability
GWAS was conducted using BOLT-LMM[37] version 2.3. LDSC[40] was used to estimate heritability and genetic correlation.

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

## Acknowledgements
The study was carried out using the UK Biobank Resource under Application Number 12006. We thank the Center for Information Technology of the University of Groningen for their support and for providing access to the Peregrine high-performance computing cluster. B.S. Tegegne is supported by a scholarship from the Graduate School of Medical Science, University of Groningen, the Netherlands.

## Author contributions
B.S.T., I.M.N., and H.S. conceptualized and designed the study. B.S.T., M.A.S., and A.M.v.R. generated the data. B.S.T., I.M.N., A.A., and M.A.S. analyzed the data. H.S., H.R., I.M.N., E.J.C.d.G., A.M.v.R., and P.v.d.H. provided critical oversight to data collection and study coordination. B.S.T., H.S., H.R., I.M.N., E.J.C.d.G., S.S., and M.A.S. wrote the manuscript. All authors contributed to interpreting the results, critical editing of the manuscript and have approved the final submission.

## Competing interests
The authors declare no competing interests.
