## [Peer Review File · Communications Biology]

Reviewers' comments:

Reviewer #1 (Remarks to the Author):

Teegne et al performed a GWAS for heart rate variability (HRV) on 46,075 individuals of European ancestry from the UK Biobank, the largest discovery GWAS to date for HRV in the European sample. Twenty-three genome-wide significant genetic variants in 16 loci were associated with HRV traits. This is the strength of the study. Another strength is the finding that individuals with lower HRV had higher risk of dying from any cause, confirmed the conclusions from previously reported studies. On the other hand, the study has some important concerns:

1. In the Methods section, authors considered that "an association was considered replicated if the direction of effect was consistent and the one-sided replication p-value was < 0.05 ". This is too relaxed and may lead to false positives. The P values from replication studies should be adjusted for multiple testing, for example Bonferroni correction ($P < 0.05/16$ or 0.003). As such, 6 loci in Table 2, including #3 (LINCR-0002), #6 (CHRM2), #8 (KCNJ5), #9 (MED13L), #13 (RGS6), and #15 (C16orf72), did not pass the correction for multiple testing ($P < 0.05/16$). Five loci needed to be excluded from the 9 loci that authors claimed to be novel.

2. Authors claimed that "the 141 genetic variants associated with low HRV combined in a GRS were not associated with mortality". The way authors computed GRS may not be correct. If the minor allele of a genomic variant is associated with low HRV, it means the common allele of that variant is associated with high HRV, and vice versa. Therefore, technically, all genetic variants significantly associated with HRV should be used for computing GRS. However, authors can select variants based on effect size, functionality and other criteria deemed appropriate.

Reviewer #2 (Remarks to the Author):

1. Supplement Figure 1:

--Please use another term instead of Caucasian.

--Journals should be phasing out this broad term. Has history in racist ideology.

2. How was European ancestry determined?

3. For the cardiovascular mortality

-- Was a competing risk model fit?

--Were non-cardiovascular mortality simply fit as censored events?

---This may bias the results.

4. Table 1: R2 BLUP

-- Apologies, what is this relationship with? Is this the R2 of the SNP with the trait (i.e. it's perfectly correlated with the trait?)

5. Did the authors attempt to see the correlation of the constructed GRS with any other traits within the UK biobank?

6. For the survival models

-- Did the authors examine the proportional hazards assumptions for the models?

-- Also whether or not there was a linear relationship with the continuous variables?

7. Can the authors walk through the advantage of analyzing both the corrected and uncorrected HRV variables?

-- If one is significant and not the other what does that really tell you? Is it anything actually biologically meaningful.

Reviewer #3 (Remarks to the Author):

The authors present results from a GWAS for HRV in UK Biobank data. They identified variants in 16 loci, and use the GWAS summary statistics to show that mortality is associated with HRV at a phenotype level, but not at a genetic level. This suggests that lower HRV does not cause higher mortality. The aim of the paper is interesting, but I have several concerns that I would like to see addressed.

1. The authors have access to study-specific and meta-analysis summary statistics for HRV as published by Nolte et al. in 2017. Therefore, it is not clear to me why the UK Biobank results were not meta-analysed with the Nolte et al. results, to maximise the sample size and statistical power to find associated loci as well as genetic associations with mortality (through a more precise GRS in UK Biobank). The conclusion from the current results is that there is no evidence of a causal role of HRV in mortality. However, with 23 associated loci, I cannot help but wonder if weak instrument bias is a concern, based on MR analyses for other exposures with a similarly low number of associated loci. The authors state that there was 100% power to find an association of mortality with the GRS if there was one, but this statement is meaningless without mentioning what effect size they had 100% power to detect (and 100% power seems a utopia).

2. I find the train of thought when the results from the GWAS are presented difficult to follow. 23 genetic variants in 16 loci; presumably the authors mean 23 independent association signals in 16 loci? The authors also state that "of the 23 genetic variants, 16 were found in the replication data set". It is not explicitly clear what "found" implies (present? significant?), nor is it described in the main paper what this replication effort consists of, and what data were used. If it's replication in the Nolte et al. meta-analysis results, then - again - it is not clear why the authors didn't simply meta-analyse the UKBB results with those previously published results in the first place, and 1) increase the power to identify associated loci; and 2) define loci as being associated if the P-value in the combined meta-analysis reaches genome-wide significance (rather than having to distinguish between identified and replicated loci). The authors further state that: "only four genetic variants could not be replicated". Is that 4 out of 23 (i.e. including the proxies), or 4 out of 16? Can the authors please clearly state how many loci they consider to be robustly associated with at least one HRV trait, how they define robust, and how many of these associations were not previously reported (based on some explicit LD criterion and/or distance criteria with previously reported results).

3. What is the difference between the 46,075 individuals (x) in the GWAS and the 54,312 individuals included only in the phenotypic association analysis? Was it absence of GWAS data in 54,312 - x individuals? Or is there another reason why those 54,312 - x were not included in the GWAS? If so, then comparing the association of mortality with HRV and GRS_HRV in two non-identical populations may not be comparing apples and apples. As a sensitivity analysis, can the authors also present results for the mortality vs. phenotypic HRV association analysis restricted only to the individuals also included in the mortality vs. genetically predicted HRV analysis?

4. Can the authors please briefly state which Nolte et al. locus was not associated with HRV in the UKBB data? Did this variant have a low MAF? Was it previously or currently genotyped or imputed? Was its association in the previous meta-analysis driven by one or very few studies? This information is useful to establish whether this locus should be discarded as a true HRV-associated locus, or whether absence of association in the UK Biobank data may have a methodological reason.

5. The authors identified several loci that were not previously associated with HRV. These loci can

provide new insights into previously unanticipated biology. Weirdly however, the results section currently focuses almost entirely on the association of mortality with (genetically predicted) HRV, while the discussion focusses to a large degree on the role of candidate genes in newly identified loci. One gene in a newly identified locus that harbours a likely deleterious variant is mentioned in the results section (NDUFA11), but is not functionally discussed in results or discussion. Three other genes first feature in the discussion section (CHRM2, GNB4, KCNJ5). Can the authors please spend some words on interesting candidate genes in newly identified loci in the results section, and state how they were prioritised? Of note, GNB4 was also previously identified by a GWAS for heart rate (den Hoed et al., 2013).

6. The authors state that an association of mortality with HRV - but not with the GRS for HRV - could reflect confounding. An additional possibility that is not yet mentioned is reverse causation. Were individuals that died soon after the ECG was acquired removed from both the phenotypic and genetic analyses to avoid bias by prevalent disease? The same question applies for the GWAS, since it may prevent identifying "HRV loci" that in reality are loci for lethal diseases that influence HRV. If individuals dying within a certain window after the ECG was obtained were indeed excluded, then what window of post-ECG death was used? Can the authors perform a sensitivity analysis with exclusion of individuals using different windows of post-ECG death to examine if results were influenced by prevalent disease? Similarly, sensitivity analysis in which only individuals are included that die after the "prevalent disease window" but within say 2, 3, 4 etc years of the ECG, then this could shed light on the extent to which HRV may be used as a biomarker for death within x years.

7. Can the authors please report the F statistic and comment on the power they had to identify a potentially true association of mortality and the GRS? Including the effect size for which they report the power.

8. Can the authors examine what causes of death other than cardiovascular mortality low HRV is associated with? E.g. cancer? And examine if cause specific mortality is associated with the GRS for HRV?

9. The genetic correlation between RMSSD and SDNN is very high (0.984). However, the difference in estimated risk of death between individuals in the 1st and 4th quartile of RMSSD and SDNN is notable (1.31 vs. 1.46). Similarly, the HR for all cause mortality with a 1 unit natural log transformed higher RMSSD (1.19) is outside the 95% CI for the effect estimate of SDNN (1.23-1.43). This suggests that the difference in risk estimate may be significantly different for the two HRV metrics, which is noteworthy. Do the authors have an explanation for this difference in effect estimate?

Minor:

10 Line 121: do the authors mean heart rate adjusted (rather than corrected)? I.e., adding heart rate as an independent variable to the model? In the discussion, the authors state that associations remain even after "considering the effect of heart rate". An effect of heart rate on mortality has not been confirmed, so please consider rephrasing to something like: "adjusting for heart rate".

Reviewers' comments:

Reviewer #1 (Remarks to the Author):

Tegegne et al performed a GWAS for heart rate variability (HRV) on 46,075 individuals of European ancestry from the UK Biobank, the largest discovery GWAS to date for HRV in the European sample. Twenty-three genome-wide significant genetic variants in 16 loci were associated with HRV traits. This is the strength of the study. Another strength is the finding that individuals with lower HRV had higher risk of dying from any cause, confirmed the conclusions from previously reported studies. On the other hand, the study has some important concerns:

We thank the reviewer for the positive remarks.

1. In the Methods section, authors considered that “an association was considered replicated if the direction of effect was consistent and the one-sided replication p-value was < 0.05 ”. This is too relaxed and may lead to false positives. The P values from replication studies should be adjusted for multiple testing, for example Bonferroni correction ($P < 0.05/16$ or 0.003). As such, 6 loci in Table 2, including #3 (LINCR-0002), #6 (CHRM2), #8 (KCNJ5), #9 (MED13L), #13 (RGS6), and #15 (C16orf72), did not pass the correction for multiple testing ($P < 0.05/16$). Five loci needed to be excluded from the 9 loci that authors claimed to be novel.

We thank the reviewer for this comment, which prompted us to re-evaluate the status of our replication evidence. We fully agree that reducing the probability of reporting false positives is very important, which is why we attempted to replicate our GWAS findings using summary statistics of our previously published GWAS meta-analysis of 20 cohorts of European ancestry in up to 28,700 participants. To accommodate your comment, we now also use and report results for a more conservative one-sided Bonferroni corrected threshold for replication of 0.00294 (corrected for the 17 independent SNPs: $0.05/17$). However, it should be noted that this replication dataset used the outdated HapMap2 as a reference panel for imputation and did not include any insertion/deletion variants (INDELs). Thus, the replication data was of lower genome-wide coverage and quality. This could only partly be compensated by our imputation of this summary statistics data against a 1000 Genome reference panel using FiZi software. We nevertheless believe that replication of our UKB GWAS findings in these data is still useful but want to emphasize that lack of replication may simply be the result of low (imputation) quality of the specific variants or SNP proxies (e.g., of the INDELs) in the replication data set. We therefore now report our replication results simply as a lookup of our GWAS findings (or proxies) in the previously published or FiZi imputed HRV summary statistics data in Supplementary Table 13. We have summarized these findings in the main text (page #4, line 88-93):

“In a lookup of the 23 association signals from the summary statistics of a previously published HRV GWAS (Nolte et al.), only 16 were present and for the remaining seven genetic variants we used proxies ($r^2 > 0.7$). Twenty one of the 23 variants replicated with a one-sided $p < 0.05$. Two genetic variants (one indel) did not reach nominal significance (one-sided $p < 0.05$). A further six variants (one indel) did not reach a more conservative one-sided Bonferroni corrected threshold of 0.00294 (0.05/17) (Supplementary Table 13).”

Given the limitations of our replication data, we believe that all of our 8 not previously reported loci are of interest. Some of these genes such as *CHRM2* and *KCNJ5* indeed provide novel biological insights (as presented in the discussion: line number 178-186 and 195-199) and have been previously reported to be associated with related traits (resting heart rate and heart rate response to exercise). However, we acknowledge that claiming these 8 as novel loci is problematic in the absence of convincing replication data and therefore, we now just state that these loci have not been reported previously. We are currently working on updating and expanding our previous GWAS meta-analysis within the *VgHRV* consortium, which will provide more definitive evidence.

2. Authors claimed that “the genetic variants associated with low HRV combined in a GRS were not associated with mortality”. The way authors computed GRS may not be correct. If the minor allele of a genomic variant is associated with low HRV, it means the common allele of that variant is associated with high HRV, and vice versa. Therefore, technically, all genetic variants significantly associated with HRV should be used for computing GRS. However, authors can select variants based on effect size, functionality and other criteria deemed appropriate.

We computed the GRSs for each HRV trait in the standard way by multiplying the effect sizes of all genome-wide significantly associated HRV genetic variants from Table 2 with their allele dosages (range 0-2) and then summing over all variants. This includes variants that initially were associated with a higher HRV, but we flipped the alleles of these variants such that the effect allele is decreasing HRV. Therefore, in line with our phenotypic analyses, we constructed weighted GRSs of HRV decreasing alleles so that lower values of the GRS reflected lower (genetically determined) HRV values expected to have adverse health effects (i.e., risk). We have now clarified this in methods (line 351-352):

“We constructed weighted GRSs of HRV decreasing alleles to evaluate the associations between genetically determined lower HRV with all-cause and cardiovascular mortality.”

Reviewer #2 (Remarks to the Author):

1. Supplement Figure 1:

--Please use another term instead of Caucasian.

We corrected in the revised submission of our flow chart ‘Caucasian’ to ‘European’ (page #1 of Supplementary figures)

--Journals should be phasing out this broad term. Has history in racist ideology.

2. How was European ancestry determined?

Now we have added *"European ancestry was defined based on self-reported ethnicity by participants"* in the method section **line #288**

3. For the cardiovascular mortality

-- Was a competing risk model fit?

--Were non-cardiovascular mortality simply fit as censored events?

---This may bias the results.

Yes, we considered non-cardiovascular mortality as censored events. However, as also suggested by reviewer 3 (point 7), we have now additionally analyzed Cancer-related mortality as a separate outcome.

4. Table 1: R2 BLUP

-- Apologies, what is this relationship with? Is this the R2 of the SNP with the trait (i.e. it's perfectly correlated with the trait?)

The replication dataset we used in this study used HapMap2 as a reference panel for imputation. To obtain data on all 1000Genomes variants, we imputed the replication dataset using FiZi (a method used to impute summary statistics) to the 1000 Genome reference. The r^2 of Best linear unbiased prediction (BLUP) of genotypes is a metric to indicate the imputation quality of the genetic variants to this 1000 Genomes reference panel calculated by FiZi. Generally, a r^2 -BLUP > 0.6 indicates a well imputed variant.

5. Did the authors attempt to see the correlation of the constructed GRS with any other traits within the UK biobank?

Thank you for pointing this out. Although we believe that such an analysis is beyond the scope of the current manuscript with its clear focus on mortality, we would like to mention that our group is currently working on a separate manuscript in which we performed a Phenome-wide association study (PheWAS) assessing the association of GRSs and genetic variants of HRV traits with 89 continuous traits and 114 diseases from the UK Biobank.

6. For the survival models

-- Did the authors examine the proportional hazards assumptions for the models?

-- Also whether or not there was a linear relationship with the continuous variables?

Yes, we took into account the model assumptions and multicollinearity was also checked. We also added the following statement in the method section **line #320-321** *"Age² was included as a covariate to account for a curvilinear relationship between HRV and age"*

7. Can the authors walk through the advantage of analyzing both the corrected and uncorrected HRV variables?

-- If one is significant and not the other what does that really tell you? Is it anything actually biologically meaningful.

Heart rate has a strong relationship with HRV, which includes a mathematical dependency of the variance on the mean inter-beat interval (IBI) that is difficult to separate from the joint vagal effects on the mean inter-beat interval (IBI) and HRV (de Geus EJC et.al. 2019). In 2016 our group introduced a parsimonious approach to correct HRV for its dependency on the mean IBI of consecutive R-peaks using coefficients of variation (Van Roon et al., 2016). The coefficient of variation detects the amount of IBI variability relative to the mean IBI of each participant. We applied this method to additionally calculate HRV values that were corrected for the influence of mean IBI. We find that associations between uncorrected HRV and morbidities/mortalities remain after correction for heart rate in the corrected HRV measures showing these associations are largely driven by HRV (mainly reflecting parasympathetic influences on the heart) rather than just heart rate (mainly reflecting influence of the sympathetic nervous system). On **page #9 line #201-202** we state: *“This finding is consistent across the different HRV traits studied and remains significant even after **adjusting** for heart rate “*

Reviewer #3 (Remarks to the Author):

The authors present results from a GWAS for HRV in UK Biobank data. They identified variants in 16 loci, and use the GWAS summary statistics to show that mortality is associated with HRV at a phenotype level, but not at a genetic level. This suggests that lower HRV does not cause higher mortality. The aim of the paper is interesting, but I have several concerns that I would like to see addressed.

We thank the reviewer for the complimentary remarks and suggestions about our paper.

1a. The authors have access to study-specific and meta-analysis summary statistics for HRV as published by Nolte et al. in 2017. Therefore, it is not clear to me why the UK Biobank results were not meta-analysed with the Nolte et al. results, to maximise the sample size and statistical power to find associated loci as well as genetic associations with mortality (through a more precise GRS in UK Biobank).

Thank you for pointing this out. We fully agree that increasing the sample size will increase power to detect novel loci. However, the GWAS summary statistics of Nolte et al. were based on imputed data using HapMap2 as the reference panel. Although we imputed these GWAS summary statistics using the 1000Genomes reference panel with FiZi, we feel that there is too much technical variation in the quality of the variants between these and the new UKB GWAS summary statistics to reliably meta-analyze them. Furthermore the main aim of our current paper was to assess the association of phenotypic HRV, and genetically predicted HRV with incident mortality and most of the other cohorts do not have data for incident mortality. We are confident that our current GRS provided us with ample power to identify a potential association with mortality (see also our reply to points 1b and 7 below).

Of note our research group is currently leading and coordinating an international consortium (*VgHRV*) that includes around 200K samples from 26 cohorts (European, Hispanic, African American and South Asian ancestry) to meta-analyze heart rate variability GWASs. In this meta-GWAS GWAS results from previous included cohorts as well as more ethnically diverse populations and new larger cohorts imputed against recent reference data such as from the Haplotype Reference Consortium will be meta-analyzed. This manuscript is under preparation, and we hope these results will help to elucidate the biological pathways through which reduced HRV contributes to cardiovascular disease risk and facilitate investigation of its potential causal role in health outcomes.

1b. The conclusion from the current results is that there is no evidence of a causal role of HRV in mortality. However, with 23 associated loci, I cannot help but wonder if weak instrument bias is a concern, based on MR analyses for other exposures with a similarly low number of associated loci. The authors state that there was 100% power to find an association of mortality with the GRS if there was one, but this statement is meaningless without mentioning what effect size they had 100% power to detect (and 100% power seems a utopia).

Initially we had similar concerns so we carefully evaluated the strengths of our GRS IVs and can assure the reviewer that weak instrument bias is not a concern based on our power calculations for which we used realistic assumptions regarding expected effect sizes. This may in fact be expected given the very large sample size of our GRS sample ($n=412,891$) and the combination of multiple genome-wide significant variants in single strong GRS IVs. About expected effect sizes, we considered hazard ratios observed in our phenotypic analyses of mortality per standard deviations of 1.86 for RMSSD and 2.2 for SDNN with all-cause mortality and 1.78 and 2.13 with cardiovascular mortality, respectively. We also considered the variance explained by the GRS for RMSSD (2.4%) and SDNN (1.4%) based on Nolte et al. (2017). Details of the power calculations are given in Supplementary Table S9. See also our reply to point 7 below for further details.

2. I find the train of thought when the results from the GWAS are presented difficult to follow. 23 genetic variants in 16 loci; presumably the authors mean 23 independent association signals in 16 loci? The authors also state that “of the 23 genetic variants, 16 were found in the replication data set”. It is not explicitly clear what “found” implies (present? significant?), nor is it described in the main paper what this replication effort consists of, and what data were used. If it's replication in the Nolte et al. meta-analysis results, then - again - it is not clear why the authors didn't simply meta-analyze the UKBB results with those previously published results in the first place, and 1) increase the power to identify associated loci; and 2) define loci as being associated if the P-value in the combined meta-analysis reaches genome-wide significance (rather than having to distinguish between identified and replicated loci). The authors further state that: “only four genetic variants could not be replicated”. Is that 4 out of 23 (i.e. including the proxies), or 4 out of 16? Can the authors please clearly state how many loci they consider to be robustly associated with at least one HRV trait, how they define robust, and how many of these associations were not previously reported (based on some explicit LD criterion

and/or distance criteria with previously reported results).

Apologies for the confusion, we revised the results section to clarify these points. We identified 23 association signals, 17 of which were independent in a total of 16 loci. We did a lookup of these signals in summary statistics of the previously published HRV GWAS by Nolte et al. to check their p-values. Of the 23 association signals, 16 were present in that summary statistics dataset and for the remaining 7 signals we used their proxies for the lookup.

We revised the result section in **line #82-88** as follows:

“..identified 23 genetic variants (17 independent) at a genome-wide significance level ($p < 5 \times 10^{-8}$). In our study, as mentioned in the method section, a locus was defined 2.5 Mb distance up- and downward of the sentinel SNP. Based on this there were 16 loci. Notably, eight of these loci (*RNF220*, *GNB4*, *LINC002*, *KLHL3/HNRNPA0*, *CHRM2*, *KCNJ5*, *MED13L*, *ALG10B*, and *C16orf72*) have not been previously reported for HRV (Table 2; Figure 1) and were not in LD ($r^2 < 0.005$) with nor within 2.5 Mb of a previously reported locus. Six previously reported loci (*RGS6*, *PPIL1*, *SYT10*, *GNG11*, *LINC00477*, and *NDUFA11*) were associated with all HRV traits (Figure 2).”

As mentioned above the replication data from the Nolte et al. meta-analysis was of lower genome-wide coverage and quality, which is one of the reasons we decided not to use this data in a meta-analysis with our UKB GWAS results. In fact, we have now downplayed the relevance of the replication data and report our replication results simply as a lookup of our GWAS findings (or proxies) in the previously published HRV summary statistics data. At the same time, as mentioned above, we are currently working on updating and expanding our previous GWAS meta-analysis within the *VgHRV* consortium, which will provide more definitive evidence. We have summarized the lookup findings in the main text (**line #88-93**):

“In a lookup of the 23 association signals in the summary statistics of a previously published HRV GWAS (Nolte et al.), only 16 were present and for the remaining seven genetic variants we used their proxies ($r^2 > 0.7$). Nineteen of the 23 variants replicated with a one-sided $p < 0.05$. Two genetic variants (one indel) did not reach nominal significance (one-sided $p < 0.05$). A further six variants (one indel) did not reach a more conservative one-sided Bonferroni corrected threshold of 0.00294 (0.05/17) (Supplementary Table 13).”

3. What is the difference between the 46,075 individuals (x) in the GWAS and the 54,312 individuals included only in the phenotypic association analysis? Was it absence of GWAS data in 54,312 - x individuals? Or is there another reason why those 54,312 - x were not included in the GWAS? If so, then comparing the association of mortality with HRV and GRS_HRV in two non-identical populations may not be comparing apples and apples. As a sensitivity analysis, can the authors also present results for the mortality vs. phenotypic HRV association analysis restricted only to the individuals also included in the mortality vs. genetically predicted HRV analysis?

The 46,075 individuals in the GWAS sample were part of the phenotypic association analyses sample (N = 54,312). However, as mentioned in the methods section and in Suppl Fig 1, we excluded around 8,000 participants in the GWAS sample due to missing/poor quality genotype, missing covariates, or if they had a diagnosis of angina, myocardial infarction, heart failure, or used anti-depressant medication, digoxin, atropine, or acetylcholinesterase that are known to influence heart rate variability. This is common practice to prevent undue confounding in GWAS analyses (a similar point is brought up by the reviewer under point 6), admittedly making the GWAS a healthier subsample of the phenotype association sample. However, most characteristics between the samples (age, BMI, resting HR, HRV, blood pressure) remained very similar as shown in Table 1. Please note that we subsequently applied GRS mortality analyses in the remaining independent sample of the UK Biobank (N=412,891) with genotype data, but who were not included in the GWAS. Participants in this remaining GRS sample did not have ECG assessments and, therefore, no HRV phenotype data available. Table 1 shows that this GRS sample was very similar to the phenotype and GWAS samples in terms of age, sex, and BMI, but had higher disease and mortality rates.

As mentioned above, unfortunately, the individuals included in the mortality vs. genetically predicted HRV analyses did not have phenotype (HRV) data. So, we could not run a sensitivity analysis for the mortality vs. phenotypic HRV association in this group.

4. Can the authors please briefly state which Nolte et al. locus was not associated with HRV in the UKBB data? Did this variant have a low MAF? Was it previously or currently genotyped or imputed? Was its association in the previous meta-analysis driven by one or very few studies? This information is useful to establish whether this locus should be discarded as a true HRV-associated locus, or whether absence of association in the UK Biobank data may have a methodological reason.

As presented in supplementary Table S6, all the reported loci (11 independent SNPs) from Nolte et al had a p-value below the Bonferroni corrected p-value ($0.05/11 = 0.00455$) and were associated with HRV in the UKBB data. We updated the text accordingly (line #106-107).

5. The authors identified several loci that were not previously associated with HRV. These loci can provide new insights into previously unanticipated biology. Weirdly however, the results section currently focuses almost entirely on the association of mortality with (genetically predicted) HRV, while the discussion focusses to a large degree on the role of candidate genes in newly identified loci. One gene in a newly identified locus that harbours a likely deleterious variant is mentioned in the results section (NDUFA11) but is not functionally discussed in results or discussion. Three other genes first feature in the discussion section (CHRM2, GNB4, KCNJ5). Can the authors please spend some words on interesting candidate genes in newly identified loci in the results section, and state how they were prioritised? Of note, GNB4 was also previously identified by a GWAS for heart rate (den Hoed et al., 2013).

We fully agree with the reviewer that identification of our HRV loci provide new insights into previously unanticipated biology. Much of the information on the biological function of reported genes was

perhaps somewhat 'hidden' in a Supplementary Note to which we now explicitly refer. We have made the following changes to create more balance between Results and Discussion sections in the revised submission:

In the result section **Line #94-105**:

"A detailed description of what is known on biological function of genes reported in this manuscript is given in the Supplementary note. ... From the newly reported genes, the lead variant (rs71784944: effect allele frequency 0.226) of the *CHRM2* locus on chromosome 7 was an indel and associated only with RMSSD. The frequency of this variant for non-Finnish Europeans in The Genome Aggregation Database (gnomAD) (<https://gnomad.broadinstitute.org/>) is 0.217. There were two lead SNPs annotated to the *KCNJ5* gene, rs7609764 which was associated with both SDNN and RMSSD and rs7102584 associated with only RMSSD. The frequency of rs7102584 is relatively low both in UKB (0.017) and in gnomAD non-Finnish Europeans (0.012) (Table 2)."

On the *NDUFA11* variant in the result section **line #121-122**:

"...This SNP had a SIFT score of 0.01 and a PolyPhen score of 0.753, suggesting deleterious effects and likely functional consequences."

In the discussion section **line #168-172**:

"The *NDUFA11* gene harbors a likely deleterious variant, which may have functional consequences. *NDUFA11* is an accessory subunit of the mitochondrial membrane respiratory chain NADH dehydrogenase complex I. In humans, a splice-site mutation in this gene is known to cause mitochondrial complex I deficiency. A recent study on mice [<https://doi.org/10.1038/s41598-018-36040-9>] has shown that downregulation of *NDUFA11* reduced ATP production and increased mitochondria reactive oxygen species production in cardiac mitochondria."

In the discussion section **line #193-194**:

"Additionally, *GNB4* has been previously reported for resting heart rate [den Hoed et al., 2013 <https://www.nature.com/articles/ng.2610>]"

6. The authors state that an association of mortality with HRV - but not with the GRS for HRV - could reflect confounding. An additional possibility that is not yet mentioned is reverse causation. Were individuals that died soon after the ECG was acquired removed from both the phenotypic and genetic analyses to avoid bias by prevalent disease? The same question applies for the GWAS, since it may prevent identifying "HRV loci" that in reality are loci for lethal diseases that influence HRV. If individuals dying within a certain window after the ECG was obtained were indeed excluded, then what window of post-ECG death was used? Can the authors perform a sensitivity analysis with exclusion of individuals using different windows of post-ECG death to examine if results were influenced by prevalent disease? Similarly, sensitivity analysis in which only individuals are included that die after the "prevalent disease window" but within say 2, 3, 4 etc years of the ECG, then

this could shed light on the extent to which HRV may be used as a biomarker for death within x years.

Thank you for raising these excellent points and suggestions. As sensitivity analyses, we ran the survival analyses after excluding individuals who died within 1 or 2 years after the ECG recording. The results show that even after excluding individuals who died within 1 or 2 years, participants with lower HRV values had a significantly higher hazard of all-cause mortality (Supplementary Table 11). Similarly, for the GWAS analyses, the effect sizes of the significant variants in the original analyses remain virtually the same after rerunning the GWAS by excluding individuals who died within 1- or 2-years post-ECG (Supplementary Table 12).

We added the following paragraph in the revised submission:

Method section, **line #383-387**:

“To minimize the influence of bias introduced by prevalent diseases, we excluded participants who died soon after the ECG recording. We repeated the phenotypic and GWAS analyses after excluding individuals who died within 1 or 2 years after ECG measurement. In addition to all-cause mortality and cardiovascular mortality, we assessed the association between HRV phenotypes and cancer mortality since cancer and cardiovascular disease are the most common causes of death.”

Result section, **line #95-98**:

“As sensitivity analyses, we reran the GWAS after excluding individuals who died within 1 or 2 years after the ECG recording. The result shows that the effect sizes of the significant variants in the original analyses remain virtually the same (Supplementary Table 12).”

Result section, **line #144-146**:

“Our sensitivity analyses show that even after excluding individuals who died within 1 or 2 years after the ECG recording, participants with lower HRV values still showed a significantly higher hazard of all-cause mortality (Supplementary Table 11).”

7. Can the authors please report the F statistic and comment on the power they had to identify a potentially true association of mortality and the GRS? Including the effect size for which they report the power.

Now we added the following statement in the method section **line #360-368**:

To ensure the strength of GRSs as IVs, we generated an *F*-statistic for each outcome. We used variance in lnRMSSD and lnSDNN explained by each set of HRV SNPs (conservatively estimated at 2.4% and 1.4%, respectively, based on Nolte et al.[ref]) to calculate the *F*-statistic using the formula $F\text{-statistic} = [R^2 \times (n - 1 - K)] / [(1 - R^2) \times K]$, where R^2 represents the proportion of variability in RMSSD and SDNN level that is explained by the GRS, n represents sample size, and K represents the number of IVs included in model (i.e., for this study $K = 1$) (Rice JA et al 1995). As a rule of thumb, an *F*-value above ten indicates that a causal estimate is unlikely to be biased due to weak instruments.

Based on these calculations the *F*-statistics for RMSSD and SDNN were 237 and 80 respectively confirming the strength of our GRS IVs.

8. Can the authors examine what causes of death other than cardiovascular mortality low HRV is associated with? E.g., cancer? And examine if cause specific mortality is associated with the GRS for HRV? To plan re-analyze for cancer with a hypothesis of negative control.

In the revised submission, we included cancer as an additional cause of death in our association analyses between HRV phenotypes and mortality.

Results section, **line #148-151**:

“For our sub-analyses for cancer as cause of death, like for all-cause mortality, participants in the lowest quartile for RMSSD and SDNN had 1.43 (95%CI:1.16-1.76; $p=8.05 \times 10^{-4}$) and 1.64 (95%CI:1.32-2.05; $p=8.94 \times 10^{-6}$) times higher risk of death from cancer compared to participants in the highest quartile (Supplementary Table 10).”

Discussion section, **line #214-219**:

“Similarly, our results show that individuals with lower HRV values had a higher risk of dying from cancer during the follow-up consistent with a systematic review by Kloter et al. in 2018 (<https://www.ncbi.nlm.nih.gov/pmc/articles/PMC5986915/>) that concluded that individuals with higher HRV and advanced coping mechanisms seem to have a better prognosis in cancer progression. Possible explanations might include that a lower HRV is associated with tumor growth through inflammation, oxidative stress, and sympathetic nerve activation.”

Methods section, Sensitivity analyses, **line #385-388**:

“In addition to all-cause mortality and cardiovascular mortality, we assessed the association between HRV phenotypes and cancer mortality since cancer was one of most common cause of death.”

9. The genetic correlation between RMSSD and SDNN is very high (0.984). However, the difference in estimated risk of death between individuals in the 1st and 4th quartile of RMSSD and SDNN is notable (1.31 vs. 1.46). Similarly, the HR for all cause mortality with a 1 unit natural log transformed higher RMSSD (1.19) is outside the 95% CI for the effect estimate of SDNN (1.23-1.43). This suggests that the difference in risk estimate may be significantly different for the two HRV metrics, which is noteworthy. Do the authors have an explanation for this difference in effect estimate?

Thank you for pointing this out. Indeed, RMSSD and SDNN are HRV indices that are highly correlated and expected to show similar effect sizes. Both sympathetic and parasympathetic activity contribute to SDNN, whereas RMSSD mainly captures short-term components of HRV that reflect parasympathetic nervous system activity on the heart. Though there was a slight difference in terms of the point estimates between SDNN and RMSSD, as presented in Figure 3, the 95% CI intervals still showed overlap (1.10-1.28 for RMSSD and 1.23-1.43 for SDNN). As such we believe the consistency and significance of effect sizes on mortality for SDNN and RMSSD is most important, and we are reluctant to overinterpret

any apparent differences in those effect sizes.

Minor:

10 Line 121: do the authors mean heart rate adjusted (rather than corrected)? I.e., adding heart rate as an independent variable to the model? In the discussion, the authors state that associations remain even after “considering the effect of heart rate”. An effect of heart rate on mortality has not been confirmed, so please consider rephrasing to something like: “adjusting for heart rate”.

Yes, we meant adjustment for mean heart rate (or Inter Beat Interval [IBI]) and have now revised this in the text. However, as mentioned in our Methods section, instead of adding heart rate as independent variable to the model, we corrected HRV for its dependency on the mean IBI of consecutive R-peaks using the coefficient of variation as suggested previously by our group (van Roon et al., 2016). The coefficient of variation detects the amount of IBI variability relative to the mean IBI of each participant. We applied this method to calculate HRV values that were corrected for the influence of mean IBI, that is the RMSSDc and SDNNc. Please also see our reply to question 7 of reviewer #2 above for details.

REVIEWERS' COMMENTS:

Reviewer #1 (Remarks to the Author):

My concerns were addressed.

Reviewer #2 (Remarks to the Author):

For this ECG sensitivity analysis, it's more that you are conditioning on surviving 1 year or 2 years. Better justification of this should be given outside of just the methods.

For the GRS survival analysis, did you repeat this in the cancer as well.

Minor

* Abstract should probably say 17 independent variants somewhere.

REVIEWERS' COMMENTS:

Reviewer #1 (Remarks to the Author):

My concerns were addressed.

Reviewer #2 (Remarks to the Author):

For this ECG sensitivity analysis, it's more that you are conditioning on surviving 1 year or 2 years. Better justification of this should be given outside of just the methods.

For the GRS survival analysis, did you repeat this in the cancer as well.

For this ECG sensitivity analysis, it's more that you are conditioning on surviving 1 year or 2 years. Better justification of this should be given outside of just the methods.

We have added the following statement in the result section on line #94-97:

‘As sensitivity analyses, we reran the GWAS after excluding individuals who died within 1 or 2 years after the ECG recording. The result shows that the effect sizes of the significant variants in the original analyses remain virtually the same indicating that our results were not biased by prevalent diseases (Supplementary Data 12).’

For the GRS survival analysis, did you repeat this in the cancer as well?

Okay, we have checked the association of HRV GRS with cancer mortality and the results are included in the revised submission. The Kaplan-Meier curves showing the association of HRV GRS with Cancer mortality are in Supplementary Figure 6.

We have included the following statement in the result section line # 155-156 ..” Similarly, genetically determined HRV was not significantly associated with cancer mortality (Supplementary Data 3 & 4, Supplementary Figure 6)”

Minor

* Abstract should probably say 17 independent variants somewhere.

We have modified the abstract “ ... In a GWAS of 46,075 European ancestry individuals from UK biobank, we identified 17 independent genome-wide significant variants...”